# Dietary and Lifestyle Changes During COVID-19 and the Subsequent Lockdowns among Polish Adults: A Cross-Sectional Online Survey PLifeCOVID-19 Study

**DOI:** 10.3390/nu12082324

**Published:** 2020-08-03

**Authors:** Magdalena Górnicka, Małgorzata Ewa Drywień, Monika A. Zielinska, Jadwiga Hamułka

**Affiliations:** Institute of Human Nutrition Sciences, Warsaw University of Life Sciences (SGGW-WULS), 159C Nowoursynowska Street, 02-787 Warsaw, Poland; magdalena_gornicka@sggw.edu.pl (M.G.); malgorzata_drywien@sggw.edu.pl (M.E.D.); monika_zielinska@sggw.edu.pl (M.A.Z.)

**Keywords:** COVID-19, coronavirus, pandemic, lockdowns, dietary behaviors, physical activity, screen time, sleeping, sociodemographics, adults

## Abstract

The present study aimed to identify patterns of dietary changes during the COVID-19 pandemic and their associations with sociodemographics, body mass index (BMI) before pandemic, and lifestyle changes in Polish adults and to examine the effects of lockdowns on dietary–lifestyle changes. This study used a cross-sectional online survey to collect data. The *k*-means algorithm was used to determine of patterns of dietary changes, and logistic regression analyses were performed. During the study period, 43% of respondents decreased physical activity (PA), 49%—increased screen time, and 34%—increased food consumption. Among the three dietary changes patterns, two opposite patterns were found: Prohealthy (28% participants) and Unhealthy (19% participants).The adherence to the Prohealthy pattern was negatively associated with age, but positively with being overweight (aOR 1.31) or obese before pandemic (aOR 1.64). Residing in a macroeconomic region with GDP > 100% decreased adherence to the Prohealthy (aOR 0.73) but increased adherence to the Unhealthy pattern (aOR 1.47). Adults over 40 years old, those living with children, unemployed, those living in a region with a higher GDP, and those not consuming homemade meals could be more exposed to unhealthy behaviors. From a public health perspective, enhancing the message “to be active” during the compulsory isolation period should be prioritized.

## 1. Introduction

Because of the worldwide spread of a novel coronavirus disease (COVID-19) on January 30, 2020, the World Health Organization (WHO) declared COVID-19 as a global pandemic. Up to June 12, 2020, nearly 8 million confirmed COVID-19 cases, including approximately 426,000 deaths, have been reported in the world, with Poland having approximately 28,577 cases and 1222 deaths [1]. Given the pandemic situation, public health recommendations and governmental measures have resulted in lockdowns and many restrictions on daily living, including isolation, social distancing, and home confinement. On 20 March 2020, the Polish government announced an epidemic status in the Republic of Poland [2], and on May 25, the government ordered social distancing, staying at home for self-isolation, remote work, and closure of preschools, schools, and universities. Gyms and swimming pools have re-opened with some restrictions on 6 June 2020.

While strict preventive measures are necessary to protect public health, they may, however, radically change individuals’ daily habits, including lifestyle-related behaviors. Staying and working at home can affect diet, food choice, and access to food and, thus, reduce possibilities and limit the practice of physical activity (PA). It was found that quarantine negatively affected the PA of the Sicilian active population, especially those of males, overweight people, and senior adults and the elderly [3]. Similarly, an international study indicated an increase in daily sitting time from 5 to 8 h per day during pandemic restrictions [4]. It should be noted that before this pandemic, insufficient PA (low PA or inactivity and excessive screen time) and obesity were described as a global public health problem [5,6]. The current COVID-19 pandemic may further worsen this situation.

In the time of COVID-19 pandemic, home fitness and the use of new technologies (videos and apps) are the solutions for being active [7]. On the other hand, because of the isolation period, avoiding sedentary behaviors or physical inactivity is difficult and, consequently, reduced PA and lower energy expenditure could negatively affect physical and mental health [8,9]. Moreover, the pandemic situation is also associated with emotions, such as fear, sadness, and anxiety, which have been indicated to reduce sleep quality [9,10]. Considering the preventive role of adequate PA as a nonpharmacological aid for health in this period [7,11] as well as the benefits of PA on psycho-physiological functions, the WHO has developed guidelines to adopt during home quarantine [12].

Furthermore, sedentary behaviors, anxiety, and boredom caused by home confinement, could influence motivation to eat, change lifestyle patterns, reduce diet quality [8], and promote overconsumption [13,14], although calorie intake should be limited with reduced PA during isolation. A pro-healthy diet, which is based on plant food (vegetables, legumes, fruits), healthy fats, and rich protein-low fat food [8,11,15], together with adequate activity is the key strategy to support the immune system and restrict seasonal and viral infections in the population [16]; thus, this strategy could support the body in fighting infections such as COVID-19 [10,16,17].

There is limited evidence to evaluate the effect of lockdowns and restrictions linked to COVID-19 pandemic on changes in dietary–lifestyle behaviors [18,19], and their results are inconsistent. Existing evidence indicated that 43.5% of Polish respondents reported eating more during quarantine, and 51.8% respondents admitted to snacking between meals more frequently [18]. In contrast, Spanish adults have adopted healthier dietary behaviors closer to the Mediterranean diet during home confinement during the COVID-19 pandemic [19]. Due to the restrictions mentioned above and the results of previous studies, we hypothesized that in this specific period, changes in dietary–lifestyle behaviors could be different depending on sociodemographics, body weight, and changes in employment and family type during lockdowns. Therefore, the present study aimed to identify dietary changes patterns and their associations with socio-demographic factors, body mass index (BMI) before pandemic, and lifestyle behavior changes. We also examined the effect of lockdowns and restrictions linked to COVID-19 pandemic on dietary–lifestyle behaviors among adults in Poland.

## 2. Materials and Methods

### 2.1. Study Design and Participants

This study was a rapid, large cross-sectional online survey. It was conducted using the Google Forms web survey platform. The link to the online survey was shared through social media, such as Facebook, Instagram, and WhatsApp, and by personal contacts of the research group members. We also asked the participants to share the study link to increase the number of persons who receive the invitation to the study and thus increase study participants. This kind of investigation allowed us to conduct a nationwide survey, especially during the rapidly changing pandemic situation in which quarantine restrictions led to limited opportunity to conduct stationary studies involving respondents. A brief description of the study and its aim and the declaration of anonymity and confidentiality were given to the participants before the start of the questionnaire. Respondents did not provide their names or contact information (including the IP address) and could finish the survey at any stage. The answers were saved only by clicking the “submit” button after filling the questionnaire.

The online survey was conducted in full agreement with the national and international regulations in compliance with the Declaration of Helsinki (2000). The personal information and data of the participants were anonymous according to the General Data Protection Regulation of the European Parliament (GDPR 679/2016). The survey did not require approval by the ethics committee because of the anonymous nature of the online survey and impossibility of tracking sensitive personal data.

We received 2575 answers; after excluding answers that met the exclusion criteria, such as living abroad, duplicate answers, pregnancy or lactation, and lack of any data, the final data set included 2381 participants (Figure 1).

### 2.2. Questionnaire

The PLifeCOVID-19 electronic survey was available between 30 April and 23 May, 2020, during the COVID-19 quarantine in Poland, which was a period of self-isolation, remote work, and prohibition of access to indoor (e.g., gyms and sports centers) and outdoor places for PA practice. This survey was open to all Poland residents aged 18 years and older. The questionnaire PLifeCOVID-19 (“Impact of the COVID-19 Pandemic on the Diet and Lifestyle of Adults”) included questions on multidimensional dietary–lifestyle changes (food group intake changes, PA, screen time, and sleeping). Reliability of the adopted questionnaire was tested through piloting, prior to survey administration. The questionnaire is provided in Appendix A.

#### 2.2.1. Dietary–lifestyle Data

The questions were related to the diet intake during the pandemic and the changes that occurred compared to the pre-pandemic period and included questions on food such as vegetables, fruits, wholegrain cereal products, low fat meat and/or eggs, pulses, fish and seafood, milk and milk products, processed meats, fast foods, salty snacks, confectionary, sweetened spreads, commercial pastry, homemade pastry, ice cream and puddings, sweetened cereals and/or cereal bars, sugar-sweetened beverages, energy drink, alcohol, water, coffee and tea, and homemade meals. Respondents were also asked about the changes in the total food intake and about difficulties in food availability For the analysis, answers were re-categorized as follows: increased intake (“I eat more”); decreased intake (“I eat less”), and no changes (answers: “I eat the same” or “I didn’t eat before and during the pandemic”).

In the next part, the questions were related to the PA, sleep, and screen time. We asked about time and self-assessment of lifestyle changes during the pandemic. -Physical activity was assessed using two questions: one on the average time spend actively and the other on change in PA during the pandemic. In the first question, respondents chose one of four categories describing their activity time which was re-categorized as follows for the analysis of this variable: low PA (<0.5 h); average PA (0.5–2 h); and high PA (>2 h). In the second question, respondents could choose one of five categories describing change in PA which were re-categorized analogically to the questions about changes in food intake for further analysis: increased PA (“my physical activity increased”); decreased PA (“my physical activity decreased”); and no changes (answers: “it has not changed, my physical activity was low before isolation and it is now the same”; “it has not changed, my physical activity was moderate before isolation and it is now the same”; “it has not changed, my physical activity was high before isolation and it is now the same”).

In questions on sleeping time, the respondents chose one of three categories describing their sleeping time or sleeping time change: “decreased,” “increased,” or “it has not changed”.

Screen time was assessed using questions on the time spend in front of the screen of a computer, TV, tablet, and/or telephone during working or non-working day as well as based on self-assessment of screen time changes during the pandemic. The respondents chose one of five categories describing their screen time, which was further re-categorized as follows: <4 h; 4–8 h; >8 h; screen time changes as: “decreased”, “increased”, “it has not changed.” If a respondent declared increased screen time, the next multiple-choice question: “Increase in the time spent in front of the screen (computer, TV, tablet, phone) is related to” concerned following the cause of it: work, entertainment, learning, boredom, or the need to help children in lessons/homework.

#### 2.2.2. Sociodemographics Data

Data on age, sex, education level, place of residence, professional situation during pandemic, and family composition during pandemic were collected.

Before the analysis, several socio-demographic variables were re-categorized. Respondents’ age were categorized into 5 categories (<30 years; 30–39 years; 40–49 years; 50–59 years; ≥60 years). Professional situation was re-categorized into three categories: (1) did not work or considerable worktime reduction (“did not work before and during pandemic”, “did not work during pandemic”, “rarely worked at the work place and did not work remotely”, “childcare”); (2) began remote work and/or study (“full-time remote working”, “part-time remote working”, “students or pupils”); (3) work in the same form as earlier (“did not change during pandemic”). Family composition during pandemic was re-categorized into 4 categories: (1) living alone (“living alone; “home-sharing”); (2) living with partner; (3) living with partner and/or children; and (4) living with parents or other relatives. Based on information on the living area (size of the place of residence and voivodeship) and the data on the percentage of EU-28 average gross domestic product (GDP) per capita [20], three macro-regions were extracted: (1) <50% of EU-28 GDP (Warmińsko-Mazurskie; Świętokrzyskie; Lubelskie; Podkarpackie; Podlaskie); (2) 50–100% of EU-28 GDP (Małopolskie; Śląskie; Wielkopolskie; Zachodniopomorskie; Lubuskie; Dolnośląskie; Opolskie; Kujawsko-Pomorskie; Pomorskie; Łódzkie; Mazowiecki regionalny); and (3) >100% of EU-28 GDP (Warszawski Stołeczny).

#### 2.2.3. Anthropometric Data

Self-reported data on height and body mass weight (last known before the pandemic) were used to calculate BMI by using the Quetelet equation (body mass (kg)/height (m^2^)) and interpreted according to the criteria of the World Health Organization [21]. Four categories were identified: underweight (BMI < 18.5 kg/m^2^), normal weight (18.5 kg/m^2^ ≤ BMI < 25.0 kg/m^2^), overweight (25.0 kg/m^2^ ≤ BMI < 30.0 kg/m^2^), and obesity (BMI ≥ 30.0 kg/m^2^) [22].

### 2.3. Statistical Analysis

In the present study, the *k*-means algorithm was used to determine dietary changes patterns during COVID-19 pandemic. Variables used to create patterns were changes of intake of vegetables, fruit, wholegrain cereal products, low fat meat and/or eggs, pulses, fish and seafood, milk and milk products, processed meats, fast foods, salty snacks, confectionary, sweetened spreads, commercial pastry, homemade pastry, ice cream and puddings, sweetened cereals and/or cereal bars, sugar sweetened beverages, energy drink, alcohol, water, coffee, and tea. Prior to analysis, all variables were re-categorized as follows: −1 (decrease of intake during pandemic); 0 (never consumed before and during pandemic; no changes in intake); and 1 (increase of intake during pandemic). Finally, three best interpretable patterns were created: (1) Prohealthy characterized by increased consumption of healthy foods, along with decreased consumption of non-recommended ones during the pandemic; (2) Constant characterized by relatively stable dietary patterns during the pandemic compared to previous time; and (3) Unhealthy characterized by increased consumption of non-recommended foods along with decreased consumption of healthy ones during the pandemic.

All the variables were analyzed qualitatively and were expressed as a percentage (%) and numbers (*n*). The chi-square test was used to analyze differences among patterns of dietary changes. Moreover, univariate and multivariate logistic regression analyses were performed to analyze the factors that influenced the odds of assignment to the (1) Prohealthy dietary changes pattern; (2) Constant dietary changes pattern; and (3) Unhealthy dietary changes pattern. The created models included sociodemographics (age; education level; family composition during pandemic; employment status during pandemic; macroeconomic region of Poland based on EU-28 average; and BMI category before pandemic) and lifestyle factors (changes in PA and screen time; and consumption of homemade meals). The results of logistic regression analyses were expressed as odds ratio (OR) or adjusted odds ratio (aOR) and 95% confidence intervals (95% CI). For all analyses, *p* ≤ 0.05 was considered significant. All analyses were performed using the Statistica 13.3 software (TIBCO Software Inc., Palo Alto, CA, USA).

## 3. Results

The sample consisted of 2381 respondents with women predominating significantly (*p* = 0.001; Table 1). Statistically, the largest group was people aged 30–39 years, and it was the highest percentage of representatives of the Unhealthy pattern (*p* < 0.001). Most respondents had a higher education, and the majority of them lived with their partner and/or children (over 50% in every dietary changes pattern); furthermore, they lived in the macroeconomic region with GDP 50–100% of EU-28 mainly (approximately 60%), regardless of the dietary changes pattern. During the pandemic, only 15% of the total respondents were employed in the same form as before. The largest percentage of them was in the Constant dietary pattern.

Nearly 70% of respondents had no difficulties with food availability during the pandemic, but 43% of respondents in the Unhealthy pattern declared such difficulties. More than half of the respondents declared no change in total food intake during the pandemic, but 64% respondents in the Unhealthy pattern ate more food.

The results demonstrated that nearly 37% of respondents in the Unhealthy pattern had low PA and only 11% had high PA (Table 2). Most of the respondents reduced their PA during the pandemic (over 40%), and approximately 65% respondents in the Unhealthy pattern had reduced PA. The highest percentage of respondents with a screen time of over 8 h on weekdays was in the Unhealthy pattern, but this group showed the lowest percentage on weekends. Approximately half of the respondents increased their screen time with the highest percentage in both the Prohealthy and Unhealthy patterns. Almost one-tenth of the respondents in the Unhealthy pattern mentioned work as a reason for increasing screen time. Over 60% of respondents slept for 6–8 h and did not change their habit, but over 30% of respondents in both Prohealthy and Unhealthy patterns increased their sleep time.

The results showed that during the pandemic, one-fifth of the total sample in the Prohealthy pattern increased the consumption of fruit, whole grain products, low-fat meat and/or eggs, and pulses. This pattern was also characterized by increased intake of vegetables and milk and milk products in approximately 30% of respondents as well as water in approximately 50% of respondents. At the same time, in the Prohealthy pattern, three-fourths of the respondents reduced their intake of fast food and commercial pastry, half of the respondents reduced intake of confectionary and salty snacks, almost one-third of the respondents reduced their intake of ice cream, and one-fifth of the respondents reduced the intake of sugar-sweetened beverages and alcohol.

The Unhealthy pattern was characterized by increased intake of processed meat, fast food, and ice cream in approximately 20% of respondents; commercial pastry and alcohol in 30% respondents; salty snacks in 50% respondents; and homemade pastry and confectionary in approximately 70% and 80% respondents, respectively. This pattern also showed the decreased intake of vegetables and fruits in approximately 60% respondents, fish in approximately 30% respondents, whole grain products in approximately 25% respondents, and water in approximately 20% respondents. The percentage of respondents consuming homemade meals increased in both the Prohealthy and Unhealthy patterns, and it was 63% and 52%, respectively.

The adherence to the Prohealthy pattern was negatively associated with age and was lower by 35% in respondents aged 40–49 years (aOR 0.65; 95% CI, 0.47–0.91), by 67% in respondents aged 50–59 years (aOR 0.33; 95% CI, 0.20–0.54), and by 78% in respondents aged at 60 years and older (aOR 0.22; 95% CI, 0.12–0.41) (Table 3) in comparison to respondents younger than 30 years. Moreover, the adherence to this pattern was lower by 27% in respondents living in the macroeconomic region with GDP >100% of EU-28 (aOR 0.73; 95% CI, 0.53–0.99) compared to those in the region with GDP < 50% of EU-28. Higher adherence to the Prohealthy pattern was shown by overweight and obese respondents (aOR 1.31; 95% CI, 1.05–1.64 and aOR 1.64; 1.19–2.27; 95% CI, respectively) compared to those with normal weight, as well as respondents with increased PA during pandemic (aOR 1.53; 95% CI, 1.18–1.98;), respondents with decrease or increase sleep time (aOR 1.50; 95% CI, 1.09–2.08 and aOR 1.36; 95% CI, 1.09–1.70), and respondents with increased consumption of homemade meals (aOR 2.03; 95% CI, 1.66–2.48), compared to those who had not changed these behaviors during the pandemic.

In the crude model, lower adherence to the Prohealthy pattern was shown by respondents with lower educational level, but higher adherence was shown by respondents living with parents or other relatives, those with remote work and/or study, and those with increased screen time (Table 3), but these associations disappeared after the adjustment.

In contrast, the adherence to the Constant pattern was positively associated with age (Table 3) and was 1.8 times higher in respondents aged 40–49 years (aOR 1.77; 1.30–2.41 95% CI), three times higher in respondents aged 50–59 years (aOR 2.99; 95% CI, 1.98–4.53), and 2.8 times higher in respondents aged at 60 years and older (aOR 2.76; 95% CI, 1.74–4.38), in comparison to respondents younger than 30 years. The adherence to this pattern was lower in respondents living with partner and/or children by 36% (aOR 0.64; 95% CI, 0.46–0.89) and by 33% in respondents living with parents or other relatives (aOR 0.67; 95% CI, 0.45–0.99) compared to those living alone. Moreover, the adherence to this pattern was lower by 30% in respondents who did not work or had considerable work time reduction (aOR 0.70; 95% CI, 0.53–0.92) and by 28% in respondents who began remote work and/or study (aOR 0.72; 95% CI, 0.55–0.96). Lower adherence to the Constant pattern was observed in obese respondents (aOR 0.58; 95% CI, 0.43–0.79) than in normal weight respondents and in respondents with decreased or increased PA during pandemic (aOR 0.60; 95% CI, 0.49–0.73; and aOR 0.69; 95% CI, 0.54–0.88, respectively), respondents with increased screen time (aOR 0.68; 95% CI, 0.56–0.82), respondents with decreased or increased sleep time (aOR 0.70; 95% CI, 0.52–0.95; and aOR 0.74; 95% CI, 0.60–0.90, respectively), and respondents with decreased or increased consumption of homemade meals (aOR 0.51; 95% CI, 0.31–0.85; and aOR 0.50; 95% CI, 0.41–0.60, respectively) compared to those who did not change these behaviors during the pandemic.

In the crude model, higher adherence to the Constant dietary changes pattern was shown by respondents with lower educational level, but lower adherences were shown by respondents with decreased screen time (Table 3); however, these associations disappeared after the adjustment.

The adherence to the Unhealthy pattern was 1.8 times higher in respondents living with partner and/or children (aOR 1.78; 95% CI, 1.17–2.72) than in those living alone, and 1.5 times higher in respondents who did not work or had considerable work time reduction (aOR 1.50; 95% CI, 1.03–2.18) than in those with work in the same form as earlier. Moreover, the adherence to this pattern was 1.4 times higher in respondents living in the macroeconomic region with GDP 50–100% of EU-28 (aOR 1.43; 95% CI, 1.04–1.99) and 1.5 times higher in respondents living in the macroeconomic region with GDP > 100% of EU-28 (aOR 1.47; 95% CI, 1.01–2.13). Higher adherence to the Unhealthy pattern was observed in respondents with decreased PA during the pandemic (aOR 2.62; 95% CI, 2.03–3.39), respondents with increased screen time (aOR 1.54; 95% CI, 1.21–1.96), and respondents with decreased consumption of homemade meals than in those who had not changed these behaviors during the pandemic.

In the crude model, higher adherence to the Unhealthy pattern was shown by respondents who began remote work and/or study, those with decreased sleep changes, and those with increased consumption of homemade meals (Table 3), but these associations disappeared after the adjustment.

## 4. Discussion

This study revealed that COVID-19 had a negative effect on physical activity (decrease in PA and increase in screen time in 43% and 49% of respondents, respectively), but the impact on food intake changes was equivocal. On the one hand, 34% of respondents declared an increase in total food intake, 33% in confectionary intake, and 18% in alcohol intake; on the other hand, 24% of respondents reported an increase in water intake, 37% showed decreased intake of fast food, and 48% showed increased consumption of homemade meals. Among the three dietary changes patterns derived with cluster analysis, two opposite patterns were found, namely, Prohealthy and Unhealthy, which represented approximately 28% and 19% of the sample, respectively. Respondents in the Prohealthy cluster more often declared moderate or high PA; an increase in the intake of vegetables, fruits, whole grains, pulses, fish, and water, and a decrease in the intake of discretionary foods during the quarantine in comparison to other respondents. Higher adherence to the Prohealthy dietary changes pattern was observed in respondents who increased their consumption of homemade meals, or increased their PA, or were overweight or obese before the pandemic. Age and living in the region with the highest GDP lowered the chance of the adherence to the Prohealthy dietary changes pattern. In contrast, a higher chance of adherence to the Unhealthy pattern was shown by respondents living with partner and/or children, those not working, and those living in a region with higher GDP. Reduction in PA, decrease in homemade meal consumption, or prolonged screen time increased the chances of adherence to the Unhealthy dietary pattern.

Our findings of reduced PA during COVID-19 pandemic are similar to the results of other studies [3,4]. Home confinement and significant restrictions reduced overall PA level and access to exercise, although more different home-based training programs appeared in media. People were unable to adapt themselves to train at home [23], and the possible causes could be lack of equipment, insufficient spaces [3], and the increase in screen time related to the beginning of remote work or children education at home. Consequently, it was more difficult for people to maintain their usual PA, but the screen time was simultaneously prolonged. Benefits of moderate and regular PA are widely described, even in the context of viral infections, as an aid to reduce inflammation and improve immune function as well as to boost immune response to viral-induced respiratory infections such as influenza or SARS-CoV-2 [24]. In contrast, inactivity can have a negative effect on physical and mental health and coping with stress and anxiety during isolation time [7]. Sedentary behavior, assessed as screen time and predominantly television viewing, is a risk factor for mortality [25], metabolic disorders, and increased risk of obesity. It should be noted that obesity was one of the most serious global problems defined as pandemic [26] before COVID-19 pandemic. Reduce in PA leads to muscle mass reduction in a few days [27], and as metabolic consequences of a prolonged sedentary lifestyle, increased body weight, including fat mass, and impairment of glycemic control are noted in healthy adults [11,28]. In overweight adults and in the elderly, the consequences are greater, which include lipid profile disturbance and increased inflammation and cardiometabolic risk [11]. Moreover, it was found that patients with metabolic disorders have a higher risk of developing severe COVID-19 infection [29,30,31]. The associations between sedentary behaviors and a higher risk of disease are largely independent of moderate to vigorous PA [25,32,33,34]. It seems that it is very important to take action to increase the awareness of health benefits resulting from an active lifestyle and to show their practical application during a different emergency condition. The message “stay at home” should not be associated with “stay on the couch”, but with “practicing more PA is better than practicing less of it” [3]. It should be noted that before the pandemic in Poland, a low percentage of people met the WHO’s recommendations for PA, and lower percentages of Polish people were actively involved in sports than those in other European Union countries [35]. Given the importance of PA in maintaining health, its promotion during a pandemic should take a new meaning. Although there were appeals for PA at home on the websites of the Ministry of Health and the National Health Fund, and fitness instructors also posted videos with exercises for the duration of the pandemic on various websites, the issue was little publicized. Therefore, it is necessary to work on special programs of gymnastics education in situations of home confinement.

Because of the risk of insufficient PA and high prevalence of overweight and obesity in Poland, this pandemic period could be critical for public health. We found that approximately 19% of respondents increased consumption of unhealthy food, and this was also associated with increased sedentary behaviors in which television viewing dominated in all population groups [36] and was associated with weight gain [32]. Our findings are partially consistent with the results of Ammar et al. [4] who showed a significant increase in sitting time and an unhealthy diet during COVID-19 outbreak, based on the data from an international online survey.

An unexpected result was higher adherence to the Prohealthy dietary changes pattern in respondents with excessive body mass. There are two possible explanations for this finding. We assume that this may be due to the greater possibility of preparing a meal at home or more free time to take care of health during home confinement. Usually, the lack of time and busy lifestyle are a barrier to incorporate dietary recommendations in daily life. In the USA and Europe, time spent on meal preparation has significantly decreased [37], but during COVID-19 pandemic, our respondents from the Prohealthy dietary changes pattern more often declared increase consumption of homemade meals, in contrast to respondents in the Unhealthy dietary changes pattern. Secondly, persons who tried to control their weight can have higher diet quality [38]. It should be noted that our results are inconsistent with those of Sidor and Rzymski [18], who found increased food consumption and snacking in Polish respondents with higher BMI. This clearly indicates the need for further research in this field.

Furthermore, we found that 53% of respondents did not change their dietary–lifestyle behaviors during the COVID-19 pandemic. Additionally, this study documents that adherence to the Prohealthy dietary changes pattern was negatively associated with age. This is compatible with the evidence that the main dietary changes patterns, healthy or unhealthy, remained consistent over time in adults of peri-retirement age [39]. Existing studies assessed dietary patterns at just one time point [40,41] and, presently, there is no research study available that describes the changes in dietary patterns during the pandemic. Generally, aging is the highest risk factor for the majority of chronic diseases and also influences the immune system, while both regular PA and prohealthy dietary habits can have positive effects on the aging immunity [42]. What the results also show is the necessity for nutrition education and scaffolding programs in nutrition and food preparation in order to deal with food and nutrition in everyday life during an emergency situation. People of different ages and lifestyle situations need competences in nutrition, food purchasing, and food preparing to maintain or to achieve a healthy lifestyle. Therefore, by conducting nutrition education, they should be encouraged and motivated to lead prohealthy lifestyle behaviors, even during emergency conditions such as a pandemic.

Because our study is the first to investigate the association between dietary changes patterns and GDP value during COVID-19 pandemic, a comparison with other results is difficult. This study showed that respondents living in the region with the highest GDP (as macroeconomic markers) have a lower chance of adherence to Prohealthy dietary changes pattern, and higher adherence to the Unhealthy dietary changes pattern. This is disturbing, because according to the results of Abbade and Dewes [43], economic development impacts on obesogenic severity by limited access to healthy food and growing urbanization. They confirmed also that industrialization and urbanization affect insufficient PA. Similarly, Zienkiewicz et al. [44] showed a strong linear relationship between the prevalence of obesity and GDP per capita in the population aged 15–29 years in Poland, but not with place of residence. Generally, urban places may be associated with an increased access to fast foods, processed foods, and consumption outside the home [45] and make an obesogenic environment, even during quarantine. It should be noted that consumers had the possibility of ordering meals home and buying food takeaways during the lockdowns. In our study, respondents in the Unhealthy dietary changes pattern more often declared difficulties with food availability (Table 1) and a decrease in the consumption of homemade meals (Table 2).

### Strengths and Limitations

The strengths of the research include, to the best of our knowledge, the fact that this is the first study on dietary lifestyle changes patterns during COVID-19 pandemic. We used an online survey, which is an ideal research tool that allowed to recruit a large sample from different regions of Poland. Furthermore, the epidemic situation makes it impossible to conduct research in such a large group of people in stationary conditions. The PLifeCOVID-19 questionnaire allowed us to collect a large amount of information related to multidimensional dietary–lifestyle changes (e.g., food group intake changes, PA, screen time, and sleeping) before and during the lockdowns. Although there are many lifestyle recommendations during pandemics, both global [15,46] and national [47], our results extend the knowledge of the need to be aware of the effect of this situation and to implement more effective interventions to modify lifestyles immediately to minimize the negative impact of COVID-19 pandemic on health in groups with higher risk.

The main limitation was the self-reported questionnaire. We decided to use this form because of specific conditions during the pandemic, as remote data collection using social networks is feasible and necessary in such cases [48]. In addition, the oversampling of a particular network due to the used method and the lack of inclusion and exclusion criteria are also limitations of the study. For example, respondents were predominantly women, with higher (university) education level. This study was also limited by the cross-sectional design of the study, which precludes the investigation of casual relationships. Moreover, because our study group was not a representative of the Polish adult population, the results of this study cannot be generalized. Finally, in this study, we considered changes in diet and lifestyle during lockdowns by self-assessment; hence, these results cannot be interpreted in the context of long-term effects.

## 5. Conclusions

Quarantine could have a bi-directional impact on dietary–lifestyle changes: positive and negative because of more time spent at home. An unhealthy effect of the COVID-19 pandemic was particularly observed in adults over 40 years of age, those with living with children, those not working, those living in a region with higher GDP, and those not consuming home meals. As the novel coronavirus is still spreading globally, which may have a lasting impact on lifestyle-related behavioral changes, from a public health perspective, it is necessary to enhance the message “to be active” during the mandatory isolation period. It is also important to define groups at a higher risk of unhealthy lifestyle-related behaviors during the COVID-19 pandemic and to create effective, targeted recommendations and tools to maintain health and to prevent other chronic diseases which exist in the population.

## Figures and Tables

**Figure 1 nutrients-12-02324-f001:**
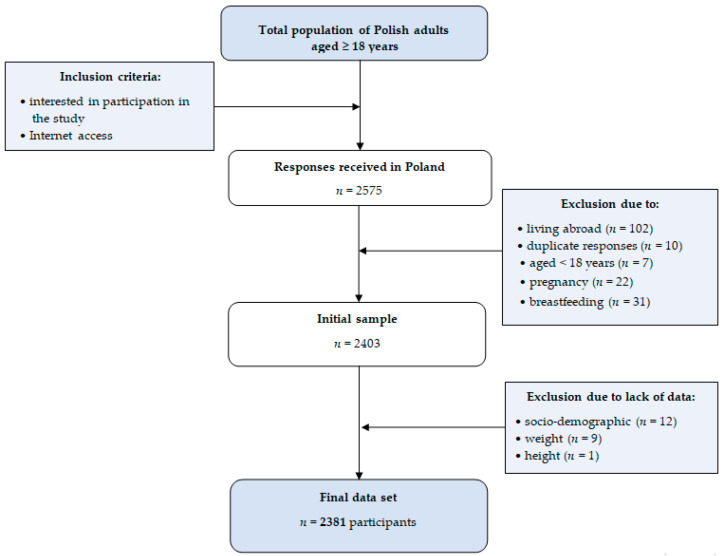
Sample collection chart.

**Table 1 nutrients-12-02324-t001:** Characteristics of respondents according to dietary changes patterns (% (*n*)).

Variables	Total 100% (*n* = 2381)	Dietary Changes Patterns			*p*-Value
		Prohealthy 27.6% (*n* = 658)	Constant 53.0% (*n* = 1262)	Unhealthy 19.4% (*n* = 461)	
**Gender:**					0.001
female	89.8 (2138)	90.6 (596)	87.9 (1109)	93.9 (433)	
male	10.2 (243)	9.4 (62)	12.1 (153)	6.1 (28)	
**Age:**					<0.001
<30 years	29.4 (700)	35.1 (231)	26.7 (337)	28.6 (132)	
30–39 years	44.8 (1067)	47.0 (309)	41.1 (519)	51.8 (239)	
40–49 years	12.9 (306)	12.2 (80)	14.3 (180)	10.0 (46)	
50–59 years	6.7 (160)	3.6 (24)	9.2 (116)	4.3 (20)	
≥60 years	6.1 (145)	2.0 (13)	8.6 (108)	5.2 (24)	
**Education Level:**					0.007
lower	22.6 (538)	19.6 (129)	25.1 (317)	20.0 (92)	
higher (university)	77.4 (1843)	80.4 (529)	74.9 (945)	80.0 (369)	
**Family Composition:**					0.039
living alone	9.9 (236)	9.0 (59)	11.2 (141)	7.8 (36)	
living with partner	22.2 (528)	21.0 (138)	23.5 (296)	20.4 (94	
living with partner and/or children	56.2 (1338)	55.9 (368)	54.8 (692)	60.3 (278)	
living with parents or other relatives	11.7 (279)	14.1 (93)	10.5 (133)	11.5 (53)	
**Place of Living:**					0.535
rural	16.2 (386)	16.3 (107)	16.4 (207)	15.6 (72)	
town <50,000 inhabitants	16.0 (382)	15.8 (104)	17.4 (219)	12.8 (59)	
town 50,000–100,000 inhabitants	11.4 (272)	12.6 (83)	11.1 (140)	10.6 (49)	
town 101,000–500,000 inhabitants	14.2 (338)	13.7 (90)	14.3 (180)	14.8 (68)	
town >500,000 inhabitants	15.8 (377)	15.2 (100)	15.2 (192)	18.4 (85)	
urban agglomeration	26.3 (626)	26.4 (174)	25.7 (324)	27.8 (128)	
**Macroeconomic Region:**					0.031
<50% of EU-28 GDP	17.0 (405)	17.8 (117)	18.4 (232)	12.1 (56)	
50–100% of EU-28 GDP	60.0 (1428)	60.2 (396)	59.2 (747)	61.8 (285)	
>100% of EU-28 GDP	23.0 (548)	22.0 (145)	22.4 (283)	26.0 (120)	
**Employment Forms During Pandemic:**					<0.001
did not work or considerable work time reduction	43.2 (1029)	40.9 (269)	43.5 (549)	45.8 (211)	
began remote work and/or study	42.0 (1000)	47.3 (311)	38.3 (483)	44.7 (206)	
work in the same form as earlier	14.8 (352)	11.9 (78)	18.2 (230)	9.5 (44)	
**BMI Before Pandemic:**					0.211
underweight	5.8 (138)	4.9 (32)	6.4 (81)	5.4 (25)	
normal weight	58.2 (1385)	56.2 (370)	58.2 (735)	60.7 (280)	
overweight	25.8 (614)	27.4 (180)	26.1 (330)	22.6 (104)	
obesity	10.2 (244)	11.6 (76)	9.2 (116)	11.3 (52)	
**Difficulties with Food Availability During Pandemic**					<0.001
no	67.8 (1615)	67.6 (445)	71.9 (907)	57.0 (263)	
yes	32.2 (766)	32.4 (213)	28.1 (355)	43.0 (198)	
**Changes in Total Food Intake During Pandemic:**					<0.001
ate less	14.1 (336)	24.8 (163)	9.8 (124)	10.6 (49)	
no changes	51.6 (1229)	41.6 (274)	66.4 (838)	25.4 (117)	
ate more	34.3 (816)	33.6 (221)	23.8 (300)	64.0 (295)	

BMI—body mass index; GDP—gross domestic product.

**Table 2 nutrients-12-02324-t002:** Lifestyle and food intake changes according to dietary changes patterns during COVID-19 pandemic (% (*n*)).

Variables	Total 100% (*n* = 2381)	Dietary Changes Patterns			*p*-Value
		Prohealthy 27.6% (*n* = 658)	Constant 53.0% (*n* = 1262)	Unhealthy 19.4% (*n* = 461)	
**Physical Activity:**					<0.001
low (<0.5 h/d)	26.1 (621)	23.9 (157)	23.4 (295)	36.7 (169)	
average (0.5–2 h/d)	57.9 (1379)	60.6 (399)	58.7 (741)	51.8 (239)	
high (>2 h)	16.0 (381)	15.5 (102)	17.9 (226)	11.5 (53)	
**Physical Activity Changes:**					<0.001
decreased	43.3 (1032)	39.4 (259)	37.6 (475)	64.6 (298)	
no changes	37.6 (895)	33.3 (219)	45.2 (571)	22.8 (105)	
increased	19.1 (454)	27.4 (180)	17.1 (216)	12.6 (58)	
**Screen Time (weekdays):**					0.021
<4 h	30.7 (7300	28.7 (189)	32.4 (409)	28.6 (132)	
4–8 h	33.5 (797)	33.4 (220)	34.6 (437)	30.4 (140)	
>8 h	35.9 (854)	37.8 (249)	33.0 (416)	41.0 (189)	
**Screen Time (weekend):**					0.031
<4 h	49.4 (1177)	47.6 (313)	52.3 (660)	44.3 (204)	
4–8 h	39.0 (929)	40.1 (264)	37.2 (470)	42.3 (195)	
>8 h	11.5 (275)	12.3 (81)	10.5 (132)	13.4 (62)	
**Screen Time Changes:**					<0.001
decreased	5.1 (122)	5.8 (38)	4.7 (59)	5.4 (25)	
no changes	45.8 (1090)	40.3 (265)	52.9 (667)	34.3 (158)	
increased	49.1 (1169)	54.0 (355)	42.5 (536)	60.3 (278)	
**Reasons of Increased Screen Time: ^1^**					
work	6.2 (148)	7.0 (46)	4.9 (62)	8.7 (40)	0.010
entertainment	4.8 (114)	4.0 (26)	5.3 (67)	4.6 (21)	0.403
learning	1.6 (38)	2.0 (13)	1.5 (19)	1.3 (6)	0.631
boredom	7.8 (186)	8.8 (58)	7.2 (91)	8.0 (37)	0.453
help children in lessons	1.0 (25)	0.3 (2)	1.3 (17)	1.3 (6)	0.087
**Sleep Per Day:**					0.103
<6 h	10.8 (256)	10.8 (71)	9.7 (123)	13.4 (62)	
6–8 h	63.0 (1501)	61.1 (402)	65.1 (822)	60.1 (277)	
≥8 h	26.2 (624)	28.1 (185)	25.1 (317)	26.5 (122)	
**Sleep Changes:**					<0.001
decreased	9.3 (221)	11.2 (74)	7.6 (96)	11.1 (51)	
no changes–low	60.7 (1445)	51.4 (338)	67.5 (852)	55.3 (255)	
increased	30.0 (715)	37.4 (246)	24.9 (314)	33.6 (155)	
**Vegetable Intake:**					<0.001
decreased	19.4 (463)	10.3 (68)	8.5 (107)	62.5 (288)	
no changes	62.1 (1478)	58.7 (386)	78.0 (984)	23.4 (108)	
increased	18.5 (440)	31.0 (204)	13.5 (171)	14.1 (65)	
**Fruit Intake:**					<0.001
decreased	20.1 (479)	13.1 (86)	9.4 (119)	59.4 (274)	
no changes	64.7 (1540)	64.4 (424)	80.3 (1014)	22.1 (102)	
increased	15.2 (362)	22.5 (148)	10.2 (129)	18.4 (85)	
**Whole Grain Product Intake:**					<0.001
decreased	11.4 (271)	10.0 (66)	6.9 (87)	25.6 (118)	
no changes	72.3 (1721)	65.8 (433)	81.4 (1027)	56.6 (261)	
increased	16.3 (389)	24.2 (159)	11.7 (148)	17.8 (82)	
**Low Fat Meat and/or Egg Intake:**					<0.001
decreased	9.7 (230)	12.3 (81)	7.1 (89)	13.0 (160)	
no changes	74.7 (1778)	66.6 (438)	81.9 (1033)	66.6 (307)	
increased	15.7 (373)	21.1 (139)	11.1 (140)	20.4 (94)	
**Pulse Intake:**					<0.001
decreased	8.5 (202)	8.7 (57)	5.3 (67)	16.9 (78)	
no changes	77.7 (1849)	70.2 (462)	84.5 (1067)	69.4 (320)	
increased	13.9 (330)	21.1 (139)	10.1 (128)	13.7 (63)	
**Fish and Seafood Intake:**					<0.001
decreased	17.0 (404)	17.6 (116)	12.1 (153)	29.3 (135)	
no changes	76.2 (1814)	72.0 (474)	82.7 (1044)	64.2 (296)	
increased	6.8 (163)	10.3 (68)	5.2 (65)	6.5 (30)	
**Milk and Milk Product Intake:**					<0.001
decreased	8.2 (195)	11.9 (78)	5.5 (70)	10.2 (47)	
no changes	71.0 (1691)	60.2 (396)	80.3 (1014)	61.0 (281)	
increased	20.8 (495)	28.0 (184)	14.1 (178)	28.9 (133)	
**Processed Meat Intake:**					<0.001
decreased	17.7 (422)	28.3 (186)	13.6 (172)	13.9 (64)	
no changes	71.4 (1699)	60.8 (400)	79.6 (1005)	63.8 (294)	
increased	10.9 (260)	10.9 (72)	6.7 (85)	22.3 (103)	
**Fast Food Intake:**					<0.001
decreased	36.6 (872)	76.4 (503)	21.3 (269)	21.7 (100)	
no changes	55.3 (1317)	19.3 (127)	72.6 (916)	59.4 (274)	
increased	8.1 (192)	4.3 (28)	6.1 (77)	18.9 (87)	
**Salty Snack Intake:**					<0.001
decreased	19.7 (469)	54.9 (361)	5.6 (71)	8.0 (37)	
no changes	62.2 (1480)	32.7 (215)	85.2 (1075)	41.2 (190)	
increased	18.1 (432)	12.5 (82)	9.2 (116)	50.8 (234)	
**Confectionary Intake:**					<0.001
decreased	18.8 (447)	51.1 (336)	6.8 (86)	5.4 (25)	
no changes	48.7 (1159)	23.1 (152)	74.3 (938)	15.0 (69)	
increased	32.5 (775)	25.8 (170)	18.9 (238)	79.6 (367)	
**Sweetened Spread Intake:**					<0.001
decreased	4.7 (111)	11.1 (73)	2.1 (26)	2.6 (12)	
no changes	91.6 (2181)	86.8 (571)	95.5 (1205)	87.9 (405)	
increased	3.7 (89)	2.1 (14)	2.5 (31)	9.5 (44)	
**Commercial Pastry Intake:**					<0.001
decreased	29.4 (701)	69.6 (458)	13.3 (168)	16.3 (75)	
no changes	59.6 (1420)	22.0 (145)	80.6 (1017)	56.0 (258)	
increased	10.9 (260)	8.4 (55)	6.1 (77)	27.8 (128)	
**Homemade Pastry Intake:**					<0.001
decreased	9.0 (215)	18.5 (122)	4.9 (62)	6.7 (31)	
no changes	51.1 (1217)	25.4 (167)	74.6 (941)	23.6 (109)	
increased	39.9 (949)	56.1 (369)	20.5 (259)	69.6 (321)	
**Ice Cream and Pudding Intake:**					<0.001
decreased	15.0 (358)	28.6 (188)	9.0 (114)	12.1 (56)	
no changes	74.9 (1784)	62.2 (409)	84.4 (1065)	67.2 (310)	
increased	10.0 (239)	9.3 (61)	6.6 (83)	20.6 (95)	
**Sweetened Cereal and/or Cereal Bar Intake:**					<0.001
decreased	6.3 (150)	14.1 (93)	3.1 (39)	3.9 (18)	
no changes	88.3 (2102)	79.6 (524)	93.5 (1180)	86.3 (398)	
increased	5.4 (129)	6.2 (41)	3.4 (43)	9.8 (45)	
**Sugar-Sweetened Beverages Intake:**					<0.001
decreased	8.4 (199)	19.6 (129)	3.7 (47)	5.0 (23)	
no changes	86.0 (2047)	74.5 (490)	93.7 (1182)	81.3 (375)	
increased	5.7 (135)	5.9 (39)	2.6 (33)	13.7 (63)	
**Energy Drink Intake:**					<0.001
decreased	5.0 (120)	8.5 (56)	3.1 (39)	5.4 (25)	
no changes	93.4 (2223)	89.4 (588)	95.8 (1209)	92.4 (426)	
increased	1.6 (38)	2.1 (14)	1.1 (14)	2.2 (10)	
**Alcohol Intake:**					<0.001
decreased	10.7 (254)	18.4 (121)	8.0 (101)	6.9 (32)	
no changes	71.3 (1697)	64.0 (421)	77.7 (981)	64.0 (295)	
increased	18.1 (430)	17.6 (116)	14.3 (180)	29.1 (134)	
**Water Intake:**					<0.001
decreased	8.8 (210)	9.0 (59)	4.9 (62)	19.3 (89)	
no changes	67.1 (1598)	43.6 (287)	80.0 (1010)	65.3 (301)	
increased	24.1 (573)	47.4 (312)	15.1 (190)	15.4 (71)	
**Consumption of Homemade Meals:**					<0.001
decreased	3.1 (75)	2.1 (14)	2.3 (29)	6.9 (32)	
no changes	48.8 (1162)	34.8 (229)	59.0 (744)	41.0 (189)	
increased	48.0 (1144)	63.1 (415)	38.7 (489)	52.1 (240)	

^1^ Multiple choice question answered by respondents who declared that their screen time increased during COVID-19 pandemic (*n* = 1169).

**Table 3 nutrients-12-02324-t003:** Associations between dietary changes patterns and sociodemographics, BMI before pandemic, or lifestyle behavior changes during COVID-19 pandemic.

Variables	Dietary Change Patterns					
	Prohealthy		Constant		Unhealthy	
	OR ^1^ (95% CI ^2^)	aOR ^3^ (95% CI)	OR (95% CI)	aOR (95% CI)	OR (95% CI)	aOR (95% CI)
**Age:**						
<30 years	1	1	1	1	1	1
30–39 years	0.83 (0.67–1.02)	0.79 (0.62–1.01)	1.02 (0.84–1.23)	1.12 (0.89–1.40)	1.24 (0.98–1.68)	1.11 (0.84–1.47)
40–49 years	0.72 (0.53–0.97) *	0.65 (0.47–0.91) **	1.54 (1.17–2.02) **	1.77 (1.30–2.41) ***	0.76 (0.53–1.10)	0.70 (0.46–1.05)
50–59 years	0.36 (0.23–0.57) **	0.33 (0.20–0.54) ***	2.84 (1.95–4.14) ***	2.99 (1.98–4.53) ***	0.61 (0.37–1.02)	0.66 (0.38–1.14)
≥60 years	0.20 (0.11–0.36) ***	0.22 (0.12–0.41) ***	3.14 (2.10–4.70) ***	2.76 (1.74–4.38) ***	0.85 (0.53–1.38)	1.00 (0.58–1.72)
**Education Level:**						
lower	0.78 (0.63–0.98) *	0.86 (0.67–1.12)	1.36 (1.12–1.66) **	1.09 (0.86–1.37)	0.82 (0.64–1.06)	1.03 (0.77–1.38)
higher (university)	1	1	1	1	1	1
**Family Composition During Pandemic:**						
living alone	1	1	1	1	1	1
living with partner	1.06 (0.75–1.51)	1.05 (0.72–1.53)	0.86 (0.63–1.17)	0.79 (0.56–1.11)	1.20 (0.79–1.83)	1.35 (0.87–2.10)
living with partner and/or children	1.14 (0.83–1.56)	1.10 (0.77–1.58)	0.72 (0.54–0.96) *	0.64 (0.46–0.89) **	1.46 (1.00–2.13)	1.78 (1.17–2.72) **
living with parents or other relatives	1.50 (1.02–2.21)*	1.16 (0.75–1.79)	0.61 (0.43–0.87)**	0.67 (0.45–0.99)*	1.30 (0.82–2.07)	1.51 (0.90–2.53)
**Employment Forms During Pandemic:**						
did not work or considerable work time reduction	1.24 (0.93–1.66)	1.18 (0.87–1.61)	0.61 (0.47–0.78) ***	0.70 (0.53–0.92) **	1.81 (1.27–2.56) ***	1.50 (1.03–2.18) *
began remote work and/or study	1.59 (1.19–2.11) **	1.17 (0.86–1.59)	0.50 (0.04–0.64) ***	0.72 (0.55–0.96) *	1.82 (1.28–2.58) ***	1.45 (1.00–2.12)
work in the same form as earlier	1	1	1	1	1	1
**Macroeconomic Region:**						
<50% of EU-28 GDP	1	1	1	1	1	1
50–100% of EU-28 GDP	1.06 (0.83–1.35)	0.93 (0.72–1.21)	0.82 (0.65–1.02)	0.85 (0.67–1.08)	1.55 (1.14–2.12) **	1.43 (1.04–1.99) *
>100% of EU-28 GDP	0.94 (0.75–1.17)	0.73 (0.53–0.99) *	0.80 (0.61–1.03)	1.02 (0.77–1.36)	1.75 (1.23–2.47) **	1.47 (1.01–2.13) *
**BMI before Pandemic:**						
underweight	0.83 (0.55–1.25)	0.71 (0.46–1.10)	1.26 (0.88–1.79)	1.43 (0.98–2.08)	0.87 (0.56–1.37)	0.87 (0.54–1.40)
normal weight	1	1	1	1	1	1
overweight	1.14 (0.92–1.40)	1.31 (1.05–1.64) **	1.03 (0.85–1.24)	0.89 (0.72–1.10)	0.80 (0.63–1.03)	0.83 (0.64–1.08)
obesity	1.24 (0.92–1.67)	1.64 (1.19–2.27) **	0.80 (0.61–1.05)	0.58 (0.43–0.79) ***	1.07 (0.77–1.49)	1.21 (0.85–1.74)
**Physical Activity Changes:**						
decreased	1.03 (0.84–1.27)	0.87 (0.69–1.09)	0.48 (0.40–0.58) ***	0.60 (0.49–0.73) ***	3.05 (2.39–3.90) ***	2.62 (2.03–3.39) ***
no changes	1	1	1	1	1	1
increased	2.03 (1.59–2.58) ***	1.53 (1.18–1.98) ***	0.51 (0.41–0.65) ***	0.69 (0.54–0.88) **	1.10 (0.78–1.55)	0.98 (0.69–1.39)
**Screen Time Changes:**						
decreased	1.41 (0.94–2.12)	1.01 (0.66–1.55)	0.59 (0.41–0.86) **	0.87 (0.58–1.29)	1.52 (0.95–2.43)	1.27 (0.77–2.09)
no changes	1	1	1	1	1	1
increased	1.36 (1.13–1.64) **	1.17 (0.94–1.44)	0.54 (0.45–0.63) ***	0.68 (0.56–0.82) ***	1.84 (1.48–2.28) ***	1.54 (1.21–1.96) ***
**Sleep Changes:**						
decreased	1.65 (1.22–2.24) ***	1.50 (1.09–2.08) **	0.53 (0.40–0.71) ***	0.70 (0.52–0.95) *	1.40 (1.00–1.97)	1.08 (0.75–1.55)
no changes	1	1	1	1	1	1
increased	1.72 (1.41–2.09) ***	1.36 (1.09–1.70) **	0.55 (0.45–0.65) ***	0.74 (0.60–0.90) **	1.29 (1.03–1.62)	1.09 (0.85–1.40)
**Consumption of Homemade Meals:**						
decreased	0.94 (0.51–1.70)	0.70 (0.38–1.30)	0.35 (0.22–0.57) ***	0.51 (0.31–0.85) **	3.83 (2.36–6.21) *	3.06 (1.81–5.17) ***
no changes	1	1	1	1	1	1
increased	2.32 (1.92–2.80) ***	2.03 (1.66–2.48) ***	0.42 (0.35–0.50) ***	0.50 (0.41–0.60) ***	1.37 (1.11–1.69) **	1.21 (0.96–1.52)

^1^ OR—odds ratio; ^2^ CI—confidence interval; ^3^ aOR—adjusted odds ratio; BMI—body mass index, GDP—gross domestic product; * *p* ≤ 0.05; ** *p* ≤ 0.01; *** *p* ≤ 0.001.

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
