# Peer review of "Dietary and Lifestyle Changes During COVID-19 and the Subsequent Lockdowns among Polish Adults: A Cross-Sectional Online Survey PLifeCOVID-19 Study"

_nutrients, 2020, doi:10.3390/nu12082324_

Round 1

Reviewer 1 Report

  1. Introduction section is too long - would recommend a shorter introduction with relevant substance.
  2. My biggest concern is the snowball sampling used for this study. In a survey like this, I suspect that snowball sampling would create significant confounding/bias. Snowball sampling is usually acceptable only to survey difficult to find or difficult to reach populations or highly specialized populations. Why did the authors choose to use snowball sampling method for this study?
  3. Was IRB approval or exemption for the study obtained? 
  4. Based on the respondents - the survey population does not seem to be demographically representative of the polish population, thus these results may not be generalizable.  

Author Response

Dear Reviewer,

Thank you for all your work on our manuscript “Dietary and Lifestyle Changes during COVID-19 and the Subsequent Lockdown among Polish Adults: A Cross-Sectional Online Survey PLifeCOVID-19 Study”. Your comments and suggestions were very useful and helped to improve the paper considerably. All your suggestions have been taken into account in the recent revision of the manuscript. You can find answers to your specific comments below.

Reviewer comments

Author responses

Reviewer 1

1.     Introduction section is too long - would recommend a shorter introduction with relevant substance.

We appreciate your valuable opinion and constructive comment. We checked the text carefully and modified our manuscript.

According to the reviewer’s comment, we have revised and shorten the introduction

2.     My biggest concern is the snowball sampling used for this study. In a survey like this, I suspect that snowball sampling would create significant confounding/bias. Snowball sampling is usually acceptable only to survey difficult to find or difficult to reach populations or highly specialized populations. Why did the authors choose to use snowball sampling method for this study?

Thank you for your comments, which have identified areas which required improvement.

We have misnamed the sampling method. It wasn't the snowball sampling - we asked participants for sharing the study link to increase the number of persons who receive the invitation to the study. We improved it in the new version of our manuscript.

It sounds like this now:

The link to the online survey was distributed via social media as Facebook, Instagram, WhatsApp and by personal contacts of the research group members. Additionally, we asked participants for sharing the study link to increase the number of persons who receive the invitation to the study and in consequence study participants. This kind of investigation allowed us to conduct a survey nationwide especially during fast changing pandemic situation which provides quarantine restrictions led to limited opportunity to conducting stationary studies involving respondents.

We hope you will find it relevant.

3.    Was IRB approval or exemption for the study obtained?

We added more details in the text.

The online survey was conducted in full agreement with the national and international regulations in compliance with the Declaration of Helsinki (2000). Personal information and data of the participants, were anonymous, according to the General Data Protection Regulation of the European Parliament (GDPR 679/2016). The survey did not require approval by the ethics committee because of the anonymous nature of the online survey and impossibility of tracking sensitive personal data.

4. Based on the respondents - the survey population does not seem to be demographically representative of the polish population, thus these results may not be generalizable

Thank you very much for pointing it out. We added comment in limitations

Reviewer 2 Report

The paper is well done and well described; results ar interesting and well described. However, the manuscript is written in poor English and therefore it must be edited by an english-speaking native in order to be published.
I would also suggest to remove the long description of the questionnaire from the "materials and methods" section, and transforming it into a table or attachment.

Author Response

Dear Reviewer,

Thank you for all your work on our manuscript “Dietary and Lifestyle Changes during COVID-19 and the Subsequent Lockdown among Polish Adults: A Cross-Sectional Online Survey PLifeCOVID-19 Study”. Your comments and suggestions were very useful and helped to improve the paper considerably. All your suggestions have been taken into account in the recent revision of the manuscript. You can find answers to your specific comments below.

Reviewer comments

Author responses

Reviewer 2

The paper is well done and well described; results ar interesting and well described. However, the manuscript is written in poor English and therefore it must be edited by an english-speaking native in order to be published.

Thank you for a thorough review. The manuscript has been corrected for language errors, using professional editing (native speaker) and proof-reading service (Translmed Publishing Group).

I would also suggest to remove the long description of the questionnaire from the "materials and methods" section, and transforming it into a table or attachment.

Thank you for your suggestion. We removed some details of the questionnaire from the "materials and methods" section, and we have included the entire Questionnaire PLifeCOVID-19 in the Appendix A.

We have corrected text to make our manuscript easy reading.

Reviewer 3 Report

Review of the manuscript nutrients-852288

Title: Dietary and Lifestyle Changes during COVID-19 and the Subsequent 
Lockdown among the Polish Adults: A Cross-Sectional Online Survey 
PLifeCOVID-19 Study

Authors: Magdalena Górnicka, MaÅ‚gorzata Ewa DrywieÅ„, Monika A. Zielinska, 
Jadwiga Hamułka

Dear Editors,

Thank you for the possibility to review the manuscript nutrients-852288.

Attached you may find my comments.

Brief summary

The manuscript presents important in-depth results in relevant categories of public health (dietary pattern and lifestyle patterns such as physical activity , screen time and sleeping habits) during a fast changing global pandemic situation with governmental quarantine restrictions due to covid-19, based on self-reported data of the respondents (Polish Adults).

The aim of the study was the identification of dietary and lifestyle behaviors and their association with socio-demographic factors, BMI before pandemic, changes in employment and family type during lockdowns. On these grounds the study is the first investigation in this field.

Broad comments

The manuscript is well-structured. All required parts are presented detailed and well-founded. The main limitation of the study (self-reported data because of lockdown restrictions) is explained properly in chapter 4 (Discussion, Strength and limitation). There are few specific recommendations to the authors, presented in the following paragraphs of the review.

Specific comments

Ad 1. Introduction:

The introduction presents a well-founded insight into governmental pandemic measures in Poland and into recent scientific results upon the effects of covid-19 pandemic measures on dietary and lifestyle patterns in other European countries (e.g. Italy and Spain), due to current studies.

From this point-of-view the presented current study in Poland is build up well-grounded.

Recommendation: Please take care of some selected linguistic comments in the introduction, e.g.:

Line 16: ‘This was by means of a cross-sectional online survey.’

Line 18-20: Please split the sentence in two sentences in order to shorten it:

‘These data revealed that the COVID-19 had a negative effect in physical activity (PA), which decreased in 43% of respondents. Screen time increased in 49% as well as food consumption which shows an increase in 34% of respondents.’

Line 32: ‘… as well as not preparing home meals…’

Line 34: ‘From a public health perspective, enhancing the message ‘to be active’ during compulsory isolation period should be prioritized.’

Line 39: Please change the sentence structure as follows: ‘Due to the spread of a novel coronavirus disease (COVID-19)  across the world, on 30 January 2020 …’

Line 40: ‘an global pandemic‘

Line 69-70: ‘…promote increased food intake and leading to overconsumption [11,12], …’

Line 77-78: ‘… which is described as having negative health effects, are linked with inflammation and metabolic disturbances [14].’

Ad 2. Materials and Methods

The method (cross-sectional online survey) is well described. Ethical principles are based on Helsinki declaration. The sample is well visualized in a sample collection chart (page 3).

Recommendation to the authors: The advantages of choosing a cross-sectional online survey (in line 109-110) may be described a bit more detailed, referring to the public situation of a global pandemic (e.g. tracking knowledge and data nationwide during a fast-moving and fast changing pandemic situation with quarantine restrictions for all people).

Materials: The electronic questionnaire with its four main parts (food intake, physical activity, screen time, sleeping habits) and with questions according to sociodemographic and anthropometric data is described very detailed. Giving examples of basic questions in between the main categories of the questionnaire (see 2.2.1, 2.2.2 and 2.2.3) is helpful, but in some parts difficult to read because of an amount of wordings in italics  (e.g. line 174-188).

Recommendation, e.g. in line 137-138: Maybe use a non-personalized wording for the description of the questionnaire, instead of:  “we asked about …”.

Statistical Analysis:

Line 205-209: Recommendation: please fill in ‘during pandemic’:

(1) Prohealthy characterized by increased consumption of pro-health foods, along with decreased consumption of not recommended ones during pandemic; (2) Constant characterized by relatively stable dietary patterns during pandemic comparing to previous time; and (3) Unhealthy characterized by increased consumption of not-recommended foods along with decreased consumption of pro-health ones during pandemic.

Line 210: All the variables was  were analyzed qualitatively

Line 211-212: Chi-square  test were was used.

Ad 3. Results                                                                                                                                                        The results are described very detailed and they are underlined with several result tables. Table 3 gives an in-depth insight in the results of the ‘associations between dietary changes patterns and socio-demographics, BMI before pandemic or lifestyle behavior changes during COVID-19 pandemic’, referring to the main aim of the study. The results of table 3 are also summarized in the following paragraphs of the manuscript (page 14) in a structured way.

Ad 4. Discussion

The discussion part compares the results of the current study with other study results (as far as available) in a well-balanced way. It also offers further research desiderates (e.g. line 383).

This part also offers  well-reflected possible explanations for some unexpected findings (e.g. line 373-383).

Recommendations for the authors:

As correctly described in line 339-365, physical activity is – also in a pandemic situation – a basic factor for individual health. Therefore it should be a part of public health efforts during  a global pandemic  situation as well (with different quarantine situations), which may be a part of peoples’ everyday life indefinitely (for a longer time). Maybe the authors could discuss some possibilities to supporting people in dealing with the changes in PA during the pandemic situation – or at least emphasize the necessity for empowering people in this.

During pandemic situation meal consumption at home increased -  for the Prohealthy dietary changes pattern an increase of homemade meals consumption is stated (line 331-333), even for the respondents with excessive body mass (line 373-376). As stated by the authors it’s very important, especially for older adults, to be encouraged and motivated to prohealthy lifestyle changes during different emergency situations such as a pandemic (line 392-394).

Maybe the authors could give a short preview as follows: What the results also show is the necessity for nutrition education and scaffolding programs in nutrition and food preparing in order to deal with food and nutrition in everyday life during an emergency situation. People of different ages and lifestyle situations need competences in nutrition, food purchasing and food preparing in order to maintain or to achieve a healthy lifestyle, especially during a public health emergency situation.

Line 336-338: Reducing PA, decreasing homemade meals consumption, or prolonging screen time has have increased the chances of the adherence to Unhealthy dietary pattern.

Line 351/352: It should be noted that obesity was/is one of the most serious global problems, defined as pandemic [26] before COVID-19.

Line 374: There are two possible explanations.

Author Response

Dear Reviewer,
Thank you for all your work on our manuscript “Dietary and Lifestyle Changes during COVID-19 and the Subsequent Lockdown among Polish Adults: A Cross-Sectional Online Survey PLifeCOVID-19 Study”. Your comments and suggestions were very useful and helped to improve the paper considerably. All your suggestions have been taken into account in the recent revision of the manuscript. You can find answers to your specific comments in the attachment.

Reviewer 4 Report

The study presented here shows dietary and lifestyle changes during the COVID-19 crisis in Polish adults. this reactive study is partly carried out during the pandemic and is carried out in a very scientific and very rigorous manner. The reactivity of the authors in their process of setting up a questionnaire and analysis of the results shows their high scientific implication.

The presentation of some data in graphical form may provide better readability of the results.

This study notably makes it possible to analyze behavioral changes in the face of such a crisis during confinement. This study seems to me to be very well conducted and therefore deserves to be published all the more since it will be a source of information for a possible second wave of Covid or any other pandemics.

Author Response

Dear Reviewer,

Thank you for all your work on our manuscript “Dietary and Lifestyle Changes during COVID-19 and the Subsequent Lockdown among Polish Adults: A Cross-Sectional Online Survey PLifeCOVID-19 Study”. Your comments and suggestions were very useful and helped to improve the paper considerably. All your suggestions have been taken into account in the recent revision of the manuscript. You can find answers to your specific comments below.

Reviewer comments

Author responses

Reviewer 4

The study presented here shows dietary and lifestyle changes during the COVID-19 crisis in Polish adults. this reactive study is partly carried out during the pandemic and is carried out in a very scientific and very rigorous manner. The reactivity of the authors in their process of setting up a questionnaire and analysis of the results shows their high scientific implication.

Thank you for reviewing our manuscript. We are honoured by the positive feedback you have given on our manuscript.

The presentation of some data in graphical form may provide better readability of the results.

We have not changed the graphic form of presenting the results due to their extensiveness. We have made every effort to make them as clear as possible.

This study notably makes it possible to analyze behavioral changes in the face of such a crisis during confinement. This study seems to me to be very well conducted and therefore deserves to be published all the more since it will be a source of information for a possible second wave of Covid or any other pandemics.

We appreciate your valuable opinion and positive comments.
